# Deciphering the Genetic Architecture of Plant Virus Resistance by GWAS, State of the Art and Potential Advances

**DOI:** 10.3390/cells10113080

**Published:** 2021-11-08

**Authors:** Severine Monnot, Henri Desaint, Tristan Mary-Huard, Laurence Moreau, Valerie Schurdi-Levraud, Nathalie Boissot

**Affiliations:** 1INRAE, Génétique et Amélioration des Fruits et Légumes (GAFL), 84143 Montfavet, France; severine.monnot@inrae.fr (S.M.); henri.desaint@inrae.fr (H.D.); 2Bayer Crop Science, Chemin de Roque Martine, 13670 Saint-Andiol, France; 3INRAE, CNRS, AgroParisTech, Génétique Quantitative et Evolution—Le Moulon, Université Paris-Saclay, Ferme du Moulon, 91190 Gif-sur-Yvette, France; tristan.mary-huard@agroparistech.fr (T.M.-H.); laurence.moreau@inrae.fr (L.M.); 4Mathématiques et Informatique Appliquées (MIA)-Paris, INRAE, AgroParisTech, Université Paris-Saclay, 75231 Paris, France; 5Université de Bordeaux, INRAE, BFP, UMR 1332, 33140 Villenave d’Ornon, France; valerie.schurdi-levraud@inrae.fr

**Keywords:** plant, virus, GWAS

## Abstract

Growing virus resistant varieties is a highly effective means to avoid yield loss due to infection by many types of virus. The challenge is to be able to detect resistance donors within plant species diversity and then quickly introduce alleles conferring resistance into elite genetic backgrounds. Until now, mainly monogenic forms of resistance with major effects have been introduced in crops. Polygenic resistance is harder to map and introduce in susceptible genetic backgrounds, but it is likely more durable. Genome wide association studies (GWAS) offer an opportunity to accelerate mapping of both monogenic and polygenic resistance, but have seldom been implemented and described in the plant–virus interaction context. Yet, all of the 48 plant–virus GWAS published so far have successfully mapped QTLs involved in plant virus resistance. In this review, we analyzed general and specific GWAS issues regarding plant virus resistance. We have identified and described several key steps throughout the GWAS pipeline, from diversity panel assembly to GWAS result analyses. Based on the 48 published articles, we analyzed the impact of each key step on the GWAS power and showcase several GWAS methods tailored to all types of viruses.

## 1. Introduction

Virus diseases are a major threat to crops. Plant infection by most virus species occurs through vectors. Climate change and globalization are prompting an expansion of the geographic area of vectors, leading to the infestation of new crop production areas [1]. Viruses are an important cause of yield loss in some crops, costing up to 100 million $ in some sectors and leading to the uprooting of million trees [2]. Some insecticide bans are currently leading to the “rediscovery” of virus damages, e.g., sudden yield losses due to the Beet yellowing virus in the French beetroot sector [3]. Fruit symptoms also contribute to harvest downgrades in vegetables or fruit trees [4]. Plant viruses are obligate and opportunistic pathogens that multiply by mobilizing the cell machinery of their hosts, they circulate from cell to cell and infect new organs through the vascular system. Plant responses to virus infection are variable, some plants are symptomatic, other display intermediate symptoms or sometimes no phenotypic changes. The sought-after phenotypes include resistance (direct halt or reduction of virus multiplication and symptom onset), tolerance (only prevention of symptom onset), and recovery (reduction of the subsequent virus load). Hereafter, we use the term ‘plant resistance’ in reference to those three phenomena. The cheapest method for farmers to avoid virus-induced crop yield loss is to grow resistant cultivars. This is more ecofriendly than conducting chemical treatments while being well adapted to a range of viruses [5,6]. Genes conferring virus resistance may be introduced in plant cultivars by bio-engineering or conventional breeding. As consumer are often reticent regarding the use of bioengineering and the technique is even forbidden in some areas worldwide such as EU [7,8], breeding is generally used for the introduction of virus resistance.

The challenge for plant breeders is to be able to detect resistance in donors in a species diversity panel and then to use them to introduce the genes conferring resistance into elite but susceptible backgrounds. As the donors are usually genetically distant from the target elite genotypes, considerable work by breeders is necessary to combine virus resistance and elite backgrounds [9]. Among potential donors, wild relatives serve as a major resistance reservoir for crops, e.g., wild lettuce (*Lactuca virosa*) is a donor of resistance to multiple viruses in lettuce breeding programs [10]. Unfortunately, the introduction of resistance from wild relative species may be difficult in some crops due to linkage drag from wild relatives that often carries unfavorable alleles that are complicated for breeders to eliminate. Moreover, crosses with cultivated species may be relatively or completely infertile [11,12]. Once resistance genes have been successfully introduced in elite cultivars, breeders may also have to cope with the arms race between their cultivars and highly mutagenic viruses, thereby reducing the time between resistance introduction and viral overcome. Finally, many crop species are simultaneously infected by several viruses [13,14]. Some viruses can join forces to overcome resistance, thus reducing the efficiency of resistant varieties in open-field conditions [15]. Polygenic resistance traits are likely more durable than monogenic resistance, but such resistance traits are more complex to understand and introduce in elite lines [16]. The challenge is therefore to be able to rapidly create resistant cultivars based on available genetic information, particularly with regard to complex and/or multiple resistance.

## 2. Mapping QTLs via GWAS Using Historical Recombinations

Genetic studies on plant virus resistance have been advancing, while following the same trends as those regarding other plant traits [17]. Studies up to the 1980s were solely focused on the description of inheritance (i.e., the number of genes putatively involved and how alleles interacted). Most early studies relied on interchromosomal genetic mixing, i.e., the random reallocation of chromosome pairs during meiosis. Genotypes whose resistance had been conferred by a single gene were used as donors in breeding programs.

Intrachromosomal genetic mixing, i.e., the exchange of some chromosome fragments by crossover during meiosis, was then taken into account to develop molecular markers segregating with the resistance phenotype. This was a key step forward. Under this scenario, the closer loci are in the genome, the higher is the probability that their alleles will come together during meiosis and be retained in the progeny [18]. Genes brought together are in linkage disequilibrium (LD), i.e., not randomly segregating, leading to more or less extended linkage blocks in the progeny. Bulked segregant analysis (BSA) is a direct use of LD to identify in a progeny issued from the cross between a resistant and a susceptible parent, whereby molecular markers that co-segregate with resistance phenotypes are identified [19]. It allowed, for example, the identification of markers linked to resistance to Citrus tristeza virus from progeny derived from an interspecific cross between *Citrus* L. (susceptible parent) and *Poncirus trifoliata* L. (resistant parent) [20]. This approach provides markers that can be used in selection, but does not aim at positioning resistance genes along the genome and is not adapted for polygenic resistances. Therefore, other approaches have been developed to position virus resistance traits into genetic maps based on studies of co-segregation between a resistance phenotype and each molecular marker of the genetic maps. For annual species, recombinant inbred lines (RILs) have become segregating populations preferentially used to map virus resistance traits, but several plant generations are required to obtain RILs. From a genetic standpoint, each RIL has a mosaic genome, each piece of which originating from the genome of one of the parents. The mapping accuracy is tightly linked to the LD, i.e., to the width of each mosaic piece (Figure 1). Each RIL can be considered as homozygous and its seed stock can be renewed if necessary, which facilitates experimentations. For example, common bean resistance to the Bean common mosaic virus was successfully mapped with a F_5_:F_6_ RIL population highlighting two loci [21]. Once a statistical association was found between markers and the resistance trait, the marker the most tightly linked to the resistance trait could be used by breeders to screen the progeny at an early stage without requiring phenotyping. Yet, as the results are partly specific to the initially crossed parents, they may not be fully extrapolated to the whole species.

Thousands genetic diversity studies have so far been focused solely on genetic differences between two parents. These differences explain why the same trait studied with RILs derived from different crosses can give different results. The different results highlight the presence of different genes or alleles among the diversity of a species controlling a same trait, or so-called genetic heterogeneity. Actually, at least 15 quantitative trait loci (QTL) have been described in rice for resistance to the Rice black-streaked dwarf virus thanks to the use of different RIL population [6]. Three cucumber cultivars whose resistances were shown to be related to three different alleles of the same VPS4-like gene were used as donors of resistance to Zucchini yellowing mosaic virus [22]. The development of multi-parent advanced generation inter cross (MAGIC) populations was the next logical step, with the aim of increasing the allelic diversity studied [23]. A MAGIC population relies on eight or more founders that are intercrossed in successive generations and are ultimately developed in RILs, i.e., a genetic mosaic of initial founder parents (Figure 1). Eight or more, rather than two parent crosses, enhance the efficient crossover rates and the allelic diversity. Using a MAGIC population, resistance to the Plum pox virus (PPV) was successfully mapped at the *A. thaliana cPGK2* locus [24].

From RILs to MAGIC populations, the diversity investigated via a segregation study was fourfold greater than in RILs or F2 progenies, but was still very limited compared to that investigated in the initial diversity panel to identify resistance donors [25,26]. Building segregating populations is quite a long and expensive process. Moreover, the homologous crossover frequency in the genomic area of interest may be low due to genetic divergence, especially if the resistance donors are from a wild relative species [27], while the extent of LD decay may be increased, and the mapping accuracy reduced.

With all these strategies, the diversity panels were investigated to discover resistance donors, and they have not been used to map traits. However, the average extent of LD decay is expected to be shorter in a diversity panel than in segregating populations; thus, using a diversity panel could potentially enhance the mapping precision [28]. Indeed, diversity panels make it possible to take advantage of all recombination events that have occurred throughout the generations (Figure 1). Unfortunately, the number of genetic markers required to cover all LD blocks in diversity panels was not affordable before the development of next-generation sequencing (NGS) technologies [6,29,30].

By decreasing the genotyping cost, NGS technologies enable genetic polymorphism characterization in large populations, with a sufficient density to cover most LD blocks. Studies of association between a phenotype and genetic markers within a diversity panel are called **genome-wide association studies (GWAS**). They rely on a set of statistical tools that have been developed for studies on humans since 2002 [31] and applied for the first time in *A. thaliana* in 2005 when deciphering the genetic determinism of flowering and plant responses to *Pseudomonas synringae* DC3000 [32]. By avoiding the segregating population building step, GWAS allows cost and time savings [29] while theoretically achieving a better resolution than linkage-based studies; hence, it is an attractive method to showcase germplasm collections [33,34,35,36]. GWAS has, however, seldom been used to map plant virus resistance (Table 1).

To our knowledge, only 48 studies have been published since 2006 (Appendix A) and plant–virus interactions were neglected in a review of GWAS for plant resistance to biotic stress [75]. Plant–virus GWAS have dealt with 16 plant species and 26 viruses, representing 32 plant–virus pairs. Only four plant–virus GWAS have been based on the *A. thaliana* model plant, probably because it is more often used in molecular biology and candidate gene focused studies. Pioneer plant–virus GWAS were focused on row crops, probably due to their economic interest, while representing 61% of the published articles (Figure 2). Among row crops, maize is the species most represented, with 13 publications. Interestingly, the number of publications in which vegetable–virus pairs were analyzed, has increased in recent years, with at least four publications in the last three years. No publications have focused on tomato, which is usually a model vegetable species. Only one publication covered a study on tree crops, which included mapping of Plum pox virus (PPV) resistance in apricot germplasm [40]. Otherwise, Potyvirus was the most investigated virus family, with eight virus species described in 15 publications. Only four GWAS considered diseases caused by co-inoculation of two viruses on the same plant. Three of these examined the Maize lethal necrotic disease (MLND) resulting from co-inoculation by the Sugarcane mosaic virus (SCMV) and Maize chlorotic mottle virus (MCMV) which cause major damage to maize crops, especially when combined [29,45,46], while the fourth study documented the effects to the Cassava mosaic disease (CMD) in cassava [71]. Four virus species were studied in more than one plant species (Table 1).

This low scientific production could have several explanations: (i) the small genomes of viruses favor studies of plant–virus interactions from the virus viewpoint, with molecular approaches preferred over genetic studies, (ii) the efficacy of monogenic resistance curbs research of complex resistance targeted by GWAS, and (iii) most accessible statistical models used in GWAS require continuous data whereas phenotyping data for resistance to viruses often are not continuous, and would require models still little-known in plant sciences (see Section 4.3.).

Like linkage map studies in segregating populations, GWAS aim to test phenotypic–genotypic variability associations, but in a diversity panel. GWAS is a shortcut term encompassing three successive steps: pre-GWAS, GWAS, and post-GWAS. The pre-GWAS step includes the diversity panel assembly, its genotyping, the analysis of its genetic diversity, and its phenotyping. Secondly, GWAS involves selecting the best statistical model and to test the marker–trait association. This results in the collection of one *p*-value per genetic marker tested, reflecting the level of the association between the marker and the resistance level. Finally, post-GWA studies consist of investigating significantly associated markers to highlight potential causal polymorphisms and assess their effects. These three steps are outlined in the following sections while focusing on points influencing statistical power of GWAS: (i) the frequency of alleles involved in resistance in the diversity panel and the effects of genes on the overall phenotype of resistance, (ii) the frequency of resistant genotypes in each genetic group within the diversity panel, (iii) the LD between resistant loci and genetic markers and (iv) the fit of GWAS models and the phenotypic distribution of the phenotype.

## 3. The Pre-GWAS Step: Assembled Diversity Panel That Is Genetically and Phenotypically Diverse

We picture the successive points illustrating a typical pre-GWA study for plant resistance to virus.

### 3.1. The GWAS Panel Composition Plays a Major Role in GWAS

GWAS on plant virus resistance have included incredibly varied **diversity panels.** Diversity panel are assembled with the aim of maximizing **the allelic diversity of** the population while limiting the population size to the maximum of the phenotyping or genotyping capacity (Figure 3a). Tamisier et al. (2017) did this by using a maximization strategy algorithm to select 310 pepper genotypes among 887, with this subset capturing 91% of the overall population allelic diversity. The number of genotypes per plant–virus GWAS population ranged from 72 apricot genotypes to 6,128 cassava genotypes [28,72]. The smallest panel highlights the difficulty of phenotyping for plant resistance to virus requiring specific inoculation conditions (PPV infection of trees, which is subject to quarantine in France) while the largest panel highlights the possibility of re-using and combining data from past experiments on a species that can be clonally multiplied and genotyped long after phenotyping. Some articles were focused on homozygous lines from germplasm collections or core collections, i.e., panels of lines pooling the maximum species diversity. The median number of genotypes in plant–virus GWAS populations was 300 (Appendix A), most of which were **homozygous lines**. Only a few plant–virus GWAS populations contained heterozygous genotypes, mainly row crops such as wheat and maize [51,58]. Markers for line panels are generally homozygous; thus, most GWAS focused on the additive effects of alleles. Heterozygous genotypes may be introduced in the diversity panel due to the biology of the species or because the species is mostly used in a hybrid context. The presence of landraces or wild relatives in a diversity panel is of major interest for the detection of resistance loci, especially if the plant virus resistance is rare or does not exist in elite lines [76]. Indeed, these genotypes are not usually grown using chemical inputs, thereby intensifying the selection pressure induced by viruses, thereby increasing the resistant allele frequencies [77]. Biological resource centers multiply genotypes via single or seed descent in populations so as to maintain their genetic diversity. In the second scenario, genotypes can therefore present residual heterozygosis and be phenotypically heterogenous. It is thus recommended to sample and bulk several plants in order to capture residual genetic segregation [78], especially if the genotype is heterogenous for the studied trait. In this case, genetic data are represented in allelic frequencies and not in allelic states.

The frequency of genetic/phenotypic variants is of major importance for plant–virus GWAS. The common disease–rare variant concept in the context of GWAS for disease resistance mapping is discussed in Kitsios et Zintzaras (2009) [79]. Rare genetic variants increase the occurrence of false positives in GWAS [80], especially rare variants from genotypes with extreme phenotypes. Rare genetic variants can result from genotyping errors leading to increased false positive rates, yet it has been reported that rare variants with major effects have not been discovered and therefore not genotyped due to low sample sizes in the concern studies [81,82,83]. To avoid false positives due to rare genetic variants, a large majority of plant–virus GWAS discarded all genetic markers with a minor allele frequency (MAF) of below 5% (Figure 3b). For instance, Xiao et al. (2019) set a 5% MAF threshold while only 59 among 1,070 rice genotypes (5.7%) were highly resistant to RBDSV [34]. The MAF threshold should instead be set according to the minor phenotype frequency within the tested population. Indeed, extreme phenotypes are usually rare in the plant virus resistance context because of emerging viruses (rare resistant phenotypes) or due to resistance already introduced via breeding (rare susceptible phenotypes) (Figure 3h). To avoid false negatives due to rare phenotypes, or so-called minor extreme phenotypes (MEP) with regards to MAF, the solution applied in human genetics has been to drastically increase the panel size [84]. GWAS populations should be enriched with genotypes from the virus ecological niche or focus on population subsets to increase the probability of detecting resistant phenotypes at adequate frequencies [28,67]. Pragmatically, Frachon et al. (2017) suggested testing different MAF thresholds and selecting the lowest one having no effect on the *p*-value distribution [85].

Finally, the presence of different genes providing the same resistance level is common in the plant virus resistance context, and should be considered when building the panel, especially for studying widespread viruses. Indeed, due to their extend area of distribution range, several isolated resistance niches may independently emerge leading to the selection of several different genes conferring the same resistance level.

### 3.2. Producing Genetic Variant Array to Enhance Positive Association Detection: A New Deal for GWAS

The power of GWAS partially relies on the capacity of available genetic markers to segregate with loci involved in resistance. **The genetic marker number** should be high enough to be sure to have at least one marker segregating with any resistance gene. This number must be adapted to the genome length and the LD extent which it is expected to decrease with the panel genetic diversity (Figure 3c,d). A panel of *A. thaliana* may require more genetic markers than a panel of maize, even if then *A. thaliana* genome is smaller than the maize one, because of the *A. thaliana* short LD extent (wild species) [86]. Likewise, autogamous species seem to require fewer markers than allogamous species [87]. Genetic marker inflation increases the selectivity of significance threshold (see Section 4.4.) [88]. Since the development of NGS technologies, single nucleotide polymorphism (SNPs) have become by far the main genetic variants used in plant–virus GWAS (42 out of 48 publications), they were used for the first time in a study on *A. thaliana* [28] and two years later in crop studies [48]. With the decline in genotyping costs, the number of genotyped accessions/lines has increased concomitantly with the number of SNPs. In maize, 2,908 SNPs were collected in 2015, while more than 1 M SNPs were collected five years later [5,46]. Row crops have the highest number of SNPs due to the complexity of some of their genomes (often polyploid) and their economic interest. Tao et al. (2013) used only 73 genetic variants, but focused on only three candidate genes in 94 maize genotypes. Nowadays at least 40% of plant–virus GWAS use genetic data derived from public databases or previous articles (Figure 4). Genetic data used in all *A. thaliana*-virus GWAS were from RegMAP (Horton et al., 2012) or the 1001 Genomes Project [28,37,38]. Once genetic data are available for a panel, they can be used to study any phenotypic trait without additional genotyping costs, provided that the diversity panel contains phenotypic diversity relevant for these new traits.

**Other genetic variants** exist, such as insertions or deletions (InDels), copy number variations (CNV), or chromosomal rearrangements. These structural variations, usually expected to be in LD with at least one SNP, have been used in only three plant–virus GWAS [35,43,65] and significant InDels were found in the two first cases. However, if the LD extent is short, the elimination of structural variants from GWAS could result in information loss. IndDels can carry as much information as SNPs: Pimenta et al. (2020) collected 35 K SNPs and 35 K InDels in a sugarcane diversity panel. Indeed, InDel polymorphisms may target genes in the accessory genome, mostly involved in biotic and abiotic stress adaptation [89], thus they are of particular interest for plant–virus GWAS. Gage et al. (2019) tested two genetic datasets to map maize resistance to the SCMV gathering either SNPs or binary code representing the expression of presence/absence variation (ePAV). The most significant SNP was located 8.5 MB away from the *ZmTrxh* gene, already validated as causal gene of the resistance. On the contrary, the most significant ePAV was directly located in *ZmTrxh*, highlighting a more accurate mapping. Structural variant studies would be especially relevant for R genes since CNV of these genes are very common, e.g., in the Vat cluster in melon [90]. K-mers are contiguous small pieces of genomic sequences extracted from ILLUMINA sequences that could be used to reveal InDels [91] and should be considered in future GWAS focused on plant–virus resistance.

Besides the number and kind of genetic variants, the reference genome from which they are derived is an emerging issue. SNPs and structural variants are determined by mapping reads of all genotypes included in the panel on **one reference genome,** and then by calling variants. Until recently, most reference genomes were assembled from short-read sequences. Many new genomes have now been released based on long-read sequencing. Obviously, the most precise genomes should be used for variant calling. For example, in cucumber, two reference genomes were assembled from the ‘Cucumber Chinese Long’ accession in 2009 and 2019, from short- and long-reads, respectively [92,93]. The latest version was enriched by 2,693 predicted genes, to reach a total of 24,317. Yet, other parameters should be considered when genomes from different genotypes are available. Gao et al. (2019) suggested that genes involved in biotic and abiotic stress adaptation could have been lost during domestication and the modern breeding process. Actually, it would probably be more interesting to use a landrace as reference genome rather than an elite line originating from intense breeding in the plant virus resistance context. The cucumber reference genome is derived from a landrace long known for its resistance to several viruses [94]. Mapping short reads of the entire panel on genomes assembled from different lines should be considered since SNP detection from a second reference genome does not require a further sequencing step. For *A. thaliana*, de novo assembly of seven genotypes highlighted the non-collinearity between the different genomes, and especially rearrangement hotspots involved in plant responses to biotic stress [95]. The reliance of GWAS on a single reference genome should therefore be questioned when looking for loci involved in plant virus resistance. Landraces and/or wild relatives aligned on a reference genome too genetically distant from them can lead to an increased amount of missing data due to misalignment leading to discard those genotypes from the diversity panel. This issue may be overcome using different reference genomes. Gage et al. (2019) used both the ‘B73’ and ‘PH207’ maize genomes. The most significant variant was located on chromosome 6 for B73 and on chromosome 0 for PH207. However, both loci were shown syntenic, corresponding to the *ZmTrxh* gene.

The reference genome concept is evolving into the **pangenome** concept, which should be considered in future plant–virus GWAS. The pangenome is the whole sequence library of a species. It consists of the core genome (including sequences shared by all individuals), the shell genome (including sequences shared by several individuals), and finally, the cloud genome (including sequences found in only one individual). Genes from the core genome are highly preserved and code for vital functions, while the accessory genome (cloud + shell) concentrates flexible genes, mostly involved in biotic and abiotic stress adaptation [89]. Gao et al. (2019) obtained a pangenome based on 586 genotypes from the *Lycopersicon* clade and compared it to the current reference genome (cv ‘Heinz1706’). They revealed that the reference genome covers up to 99.6% of the core genome, but only a third of the accessory genome. Pangenome subsets can also be focused on specific gene families involved in virus resistance, such as nucleotide-binding sites leucine-rich repeat (NBS-LRR) immune receptors. In *A. thaliana*, cloud and shell NLRomes were respectively found to pool 3,932 and 1,495 NLRs among the 11,497 detected NLRs and only half of the NLRs were shared by the 60 genotypes studied [96]. Single copy regions retrieved from independent assemblies of several lines could be gathered in a single-copy pangenome that could be used as a reference to derive biallelic structural variations [91]. GWAS toolboxes are now being specifically developed to use genetic data derived from pangenomes. However, from a GWA viewpoint, non-collinearity between two genomes due to insertion, deletion, and inversion can result in the emergence, loss, or shift of QTLs involved in plant virus resistance [97]. In addition, non-collinearity reduces the recombination rate, thereby favoring mapping by GWAS instead of RIL analysis [95]. The use of a pangenome instead of different reference genomes would help combine all information and make it possible to perform a single GWAS instead of several and to compare the results. Multiple reference genomes and pangenomes are very promising for the potential discovery of new QTLs controlling plant virus resistance. However, this strategy could be prohibitively expensive for species with very large genomes. In such cases, an alternative/complementary strategy would be to target diversity only in exome genes or especially in resistance genes via R gene enrichment sequencing by sequence capture, as has been carried out on wheat to identify stem rust resistance genes by GWAS [98].

### 3.3. Examining Resistance Distributions among Genetic Groups of the Panel

Long range LD between physically distant genetic variants is an outcome of the genotype history, and more specifically from the fact that panels can seldom be considered as a panmictic population, but is composed of individuals originated from different (often unknown) groups of related individuals (Figure 3f,g). This genetic structure may result in false positive if the groups showed contrasted resistance levels (Figure 3l). In theory, the genetic structure is not problematic for GWAS when the plant and virus did not coevolve, e.g., when the plant is not a natural host of the virus. Pagny et al. (2012) did not detect any correlations between the geographical origin of the wild species *A. thaliana* and PPV resistance. Otherwise, this genetic structure should be considered in the GWAS model since the unbalanced distribution of resistance alleles among genetic groups is widely reported in plant disease studies. An increased resistant allele frequency may occur in ecological niches where viruses and plants have coexisted and where alleles conferring resistance are expected to provide a fitness gain that is greater than its energy cost [77]. If a genetic group is particularly resistant to the virus studied, all loci isolating this population subset from the rest of the panel will be associated with the resistance phenotype regardless of their vicinity to loci involved in resistance.

LD can also be related to the recent history of inbred lines in the panel. Indeed, the resistance allele frequency might be closely correlated with the market segmentation of the species. For instance, within the same species, lines targeted for cropping in open or protected fields will not have the same virus resistance requirements. Breeders therefore select more or less intensively for a specific type of resistance, thereby generating co-segregation between the phenotypic and genetic structure. Moreover, breeding may sometimes retain two elite genotypes that are almost clones and cannot be considered as being independent genotypes, impairing results if their relatedness is not accounted for in the detection model. The genetic proximity between lines can be estimated by pairwise kinship coefficient calculation [99] (Figure 3e).

Hence, to prevent the detection of false-positive, the majority of plant–virus GWAS used models accounting for both the panel structure and the kinship between individuals (see Section 4.2.). Another way to overcome unbalanced resistance frequencies among groups is to create population subsets and focus a GWAS on every genetic group, if the groups pool enough genotypes [34,72]. For instance, the maize-MNLD GWAS focused on the whole population or on two groups did not map the same QTLs [45]. Thus, if resistance is only detected in the genotypes of a wild relative species, GWAS should be carried out on this species if enough genotypes are available [86]. This kind of panel often has a short LD extent, which provides better resolution for QTL mapping.

### 3.4. Phenotyping: Tricky Steps for a Powerful GWAS

The phenotyping quality may be quantified via the broad-sense heritability or experimental repeatability, calculated as:(1)$$H^{2}=\frac{\sigma_{G}^{2}}{\sigma_{G}^{2}+\frac{{\;\sigma }_{\epsilon }^{2}}{\mathrm{rep}}}$$
where $\sigma_{G}^{2}$ is the genotypic variance, $\sigma_{\epsilon }^{2}$ is the environmental variance, and rep is the number of plant repetition per genotype.

$H^{2}$is the part of the phenotypic variance explained by the genetic variance. The power of GWAS is highly dependent on the capacity of the phenotyping protocol to accurately reflect the plant response, and thus, is improved by a high H^2^ (Figure 3k). We will use this indicator throughout this section to analyze the phenotyping efficiency in plant–virus GWAS.

The first step to phenotype plant virus resistance is obviously virus inoculation. **Natural**
**inoculation**—via vectors in open field trials—has been the main chosen strategy in GWAS so far (Figure 5). The challenge with natural inoculation is to be sure that the whole trial has been visited by viruliferous vectors. It is hard to distinguish between symptomless resistant plants and non-inoculated ones. A few tips have been proposed to increase the chance of having a sufficient viral pressure in open field trials, including: find a relevant location, reproduce experiments across years, and set up an experimental design that allows statistical correction for field heterogeneity in phenotypes. Wolfe et al. (2016) pooled datasets from five populations to study cassava resistance to CMD. The five populations were tested at several locations during different years and the phenotypes were corrected with a statistical model taking environmental effects into account.

Many viruses can be **artificially inoculated** in plants, by hand rubbing, grafting, agrobacterium-inoculating, or even via mist blowing (Figure 5). Hand rubbing, the technique most frequently implemented in plant–virus GWAS, enables homogenization of the inoculum concentration, therefore enhancing the experimental reproducibility. Only three articles regarding studies using mechanical inoculation were focused on more than one strain. Co-inoculation can be mimicked by artificial inoculation, but has been seldom used. Nyaga et al. (2020) used a mist blower to inoculate MCMV and SCMV in a maize field (1:4 concentration). Grafting, i.e., an alternative when hand rubbing is inefficient, was used by Mariette et al. (2016) to inoculate PPV in an apricot diversity panel. Grafting and hand rubbing inject more virus than natural inoculation [100], but the impact of this higher concentration on resistance mapping has not been studied. When inoculation by rubbing or grafting is not possible, controlled vectorial inoculation can be set up. This requires a large and genetically stable vector rearing scheme. Indeed, mass rearing can be quite challenging, especially when dealing with species with sexual reproduction that poorly tolerate inbreeding or those with winged and/or highly mobile adults. This strategy has only been adopted in four plant–virus GWAS for both the Barley yellow dwarf virus (BYDV) and the Sugarcane yellow leaf virus which are two persistent viruses [30,48,55,65]. Like other artificial inoculations, vectorized inoculations do not perfectly reflect natural conditions. Choudhury et al. (2019) tested a wheat diversity panel for resistance to BYDV with both vectorized and natural inoculation protocols and did not observe clear correlations between the resulting phenotypes. In natural conditions, vectors can choose to colonize the most susceptible variety within the population. Vectorized inoculation enables researchers to assess the vector impact in the resistance process, including comparison with the artificial inoculation result if this latter technique can also be implemented. This strategy was used to study the inheritance of the resistance to begomoviruses/*B. tabaci* in tomato [13], but has not been implemented in any GWAS to date, possibly because this would require two large series of experiments in addition to vector rearing.

To compare experimental heritabilities regarding the different inoculation methods, we select 47 experiments conducted in 12 plant–virus GWAS in which the ‘symptom severity’ was scored (including 38 experiments scored from 1 to 5). Only two studies concerned a vectorized virus and exhibited contrasted heritabilities. Twelve experiments concerned mechanically inoculated viruses and 33 natural infections. Unexpectedly, on average heritability was equivalent when virus symptoms were scored after natural infection and after mechanical inoculation (Figure 6). However, a very broad range of heritabilities of cassava symptoms caused by CBSV was noted (0.11 to 0.73) according to the locations and years [73]. Barley resistance to the Barley mild mosaic virus (BaMMV) and to the Barley yellow mosaic virus (BaYMV) were studied with both mechanical and natural inoculation. The heritabilities were significantly higher with the former method for the BaMMV (0.69 and 0.39, respectively), while there was no difference for the BaYMV (0.42 and 0.41) [60].

Symptom scoring, which reflects the whole **plant response**, is fast, cheap, and non-destructive, but is the work of experts [101]. However, symptom scoring has one main drawback, i.e., it does not guarantee that the symptoms observed are caused by the studied virus, and therefore, it could be inefficient in scoring natural inoculation experiments if several virus species are involved. The presence/absence of symptoms is the simplest scoring strategy and is comparable to case/control studies in human medicine. It provides binary phenotypic data and has been only used in nine plant–virus GWAS, mostly to calculate the incidence. Plant responses can be scored with increasing series reflecting the symptom severity and with so-called ordinal classes (Figure 3h). Indeed, a plant scored 4 is not twofold more resistant than a plant scored 2, these discontinuous data may require further statistical transformation or specific GWAS models to be suitable for study (see Section 4.1.). This is the most popular resistance scoring method and has been used in at least 28 plant–virus GWAS. Quantitative traits, such as the proportion of impacted organs or the extent of differences in physiological traits (dry weight, flowering time, etc.) between infected and healthy plants, can also accurately reflect the plant response level. Thus, delayed growth caused by viral infection was used by Tamisier et al. (2020) to map pepper tolerance to the Potato virus Y (PVY). Based on the genotype reproducibility, the phenotypic data of several repetitions may be combined to obtain a new phenotype at the genotype level. For instance, the proportion of symptomatic plants reflects the virus incidence and was scored 17 times in plant–virus GWAS. Combining the presence/absence of symptom data and symptom severity gives a more complex resistance phenotype, i.e., the disease severity index, and was used by Hao et al. (2015).

**Virus detection** is aimed at reflecting the capacity of a virus to complete its cycle while mirroring the plant response scoring. Enzyme-linked immunosorbent assay (ELISA) and polymerase chain reaction (PCR) tests have been widely used to detect viruses, they respectively rely on the complementarity between antibodies and virus proteins [102] and (RNA) DNA amplification of virus parts [103]. Plant samples are collected, shredded and analyzed in the laboratory (Figure 3i). Because of this heavy process, populations phenotyped by ELISA were understandably smaller than those phenotyped by symptom scoring, i.e., seldom exceeding 200 genotypes. ELISA and reverse transcription quantitative PCR (RT-qPCR) can produce both qualitative and quantitative data, but quantitative data are more expensive and time consuming than qualitative ones. Four plant–virus GWAS used quantitative ELISA and three plant–virus GWAS used (RT)-qPCR to measure the Cucumber mosaic virus (CMV) and the Sugarcane yellow leaf virus accumulations in an *A. thaliana* and a sugarcane diversity panel, respectively [39,65]. Virus accumulation calculated by ELISA or (RT)-qPCR as phenotypic trait gave high heritabilities of between 0.5 and 0.75.

New methods enable in-plant virus monitoring and may give new types of phenotypic data. Two techniques are currently used: the immune-fluorescence technique where a fluorescence protein (FP) tagged antibody targets the virus, and the fluorescence technique, where an FP gene is directly introduced in the virus sequence. In this latter technique, viruses may be tracked via the transcription of transformed virus sequences (including FP introduced sequences) by the plant cell machinery. To our knowledge, this technique was implemented in two published studies, using a mutant PVY-GFP strain to assess the number of primary foci on cotyledons, with excellent experimental heritability of up to 0.98 [42,43]. Zhang et al. (2020) used a Turnip mosaic virus (TuMV) strain tagged with a green FP (GFP), but phenotyped the panel with symptom severity scoring and (RT)-qPCR. Fluorescence techniques are not yet 100% reliable. First, leaf overlap can hamper measurement of the infected surface. Second, viruses with shorter sequences may be favored, hence those having lost the GFP sequence. In such plants, an infection may increase without any fluorescence observed. Conversely, GFP can sometimes freely migrate in plants without viruses. Virus transformation is the time consuming and costly step, but once the mutant virus is available, a diversity panel can be tested without excessive increased cost. By using two fluorescent proteins of different colors (GFP, YFP, etc.), this technique could also be used to study virus co-inoculated. However, the FP method is limited to certified labs and controlled conditions.

Both symptom scoring and virus detection could be performed several times post-inoculation. This is clearly an opportunity to describe resistance pattern in the same plant or the kinetics of infection in a set of plants. Infection kinetics measured in each genotype through repetitions could reflect the resistance phenotype, including: the area under the disease progress curve (AUDPC) [104], AUDPC stairs (AUDPS) [105], or Gompertz curves (Schoeny et al. 2017). All obtained data are continuous, but only eight plant–virus GWAS have used AUDPC, five of them scoring maize and cassava and the last one with *A. thaliana*. For this phenotype, the plant must be grown a long enough time for scoring, which may be crippling for non-row crops, and the scoring repetitions may be too heavy to implement in a large panel of plants.

## 4. The GWAS Step: Choosing a Model Tailored for the Datasets

Historically, GWAS involve a succession of N runs of a statistical model that tests associations between resistance variations and N genetic variants, e.g., when 10,000 SNPs are available, the model is run 10,000 times. GWAS results consist of a collection of *p*-values of the different markers (10,000 *p*-values when 10,000 SNPs are available), reflecting the quality of association between each genetic variant and the resistance level. Different statistical models have been used in plant–virus GWAS, such as Chi2, generalized linear models (GLM), and linear mixed models (LMM). The challenge is to select the statistical model that fits the phenotypic data. Indeed, resistance to virus phenotypic distributions seldom follow a typical Gaussian distribution of quantitative traits studied by GWAS (see Section 3.4.). Recently, new GWAS models closely tailored to resistance biology (multilocus, dominance interactions, plant/virus diversity interactions, etc.) have been published and are presented in the following sections and illustrated in Figure 7.

### 4.1. Linear Mixed Models (LMM) Are the Most Used in GWAS

LMMs are the most popular statistical models used in plant–virus GWAS, with 42 out of the 48 resistances studied using LMM, or a derivative LMM. Yu et al. (2006) described an LMM that jointly corrects for the population genetic structure (fixed effect) and kinship (random effect):(2)$$Y=\;\mu +{\;X}_{M}\beta +X_{W}\nu +g+\;e$$
where $g\;\to N\left(0,{K\sigma }_{g}^{2}\right)$ and $e\;\to N\left(0,{I\sigma }_{e}^{2}\right)$ g, e independent

Y is the phenotype vector;

µ is the vector of intercept;

$X_{M}$ is the incidence matrix of the allelic state at one marker;

β is the vector of allelic effect;

$X_{W}$ is the incidence matrix of the fixed effects, including the population genetic structure;

ν is the fixed effect vector;

g is the vector of polygenic effects (random);

K is the kinship matrix between individuals and $\sigma_{g}^{2}$ is the genetic variance; and

e is the vector of residuals and $\sigma_{e}^{2}$ is the residual variance

In plant–virus GWAS, five MLM GWAS did not correct for the genetic structure or kinship, 16 GWAS corrected only for kinship, 13 GWAS corrected only for the genetic structure, and 33 GWAS corrected for both. Only 13 plant–virus GWAS articles tested more than one model. The importance of kinship and structure in plant–virus GWAS is described in the previous section (see Section 3.3.).

Population structure and kinship are calculated from the same genetic dataset, with many different tools available to calculate them. For instance, the population structure can be investigated by multivariate analysis such as principal component analysis (PCA) or discriminant analysis of principal components (DAPC) [106] (Figure 3f), or Bayesian analysis implemented in STRUCTURE [107] (Figure 3g). Methods that are faster and more adapted to handle large datasets were recently made available, e.g., SNMF [108] and ADMIXTURE [109]. All methods calculate the percentage affiliation of each line to the underlying groups, and the tricky point is to determine how many groups are present in the diversity panel. Kinship estimates the degree of genetic covariance among individuals [110]. There are many ways to calculate kinship coefficients, all of which use SNP subsets that are randomly chosen, or with an MAF where rare alleles are considered as genotyping errors. A common approach is to use the estimator described by Van Raden (2008) [99] (Figure 3e). It has been shown that the detection power was lower for markers located in regions with strong LD when these markers were used to compute the kinship matrix. To avoid this phenomenon (so-called proximal contamination) and improve the GWAS power [111], an alternative among others is to calculate n kinships containing SNPs of n-1 chromosomes, and to correct each GWAS run with a kinship calculated without SNPs from the chromosome being tested. Sometimes, correcting for both the genetic structure and kinship can increase the rate of false negatives. In this case, the kinship matrix captures part of the genetic structure of the population and is sufficient for GWAS.

Under the null hypothesis (no QTLs associated with the phenotype), the distribution of *p*-values collected in the GWAS step should have a uniform distribution. As this hypothesis is expected to be true for most variants, the QQplot, representing the quantile of empirical *p*-values, versus the theoretical quantile, should be a bisector with only a few points (corresponding to variants with low *p*-values) diverging from it in the higher quantile (Figure 7b). Twenty-three among 48 plant–virus GWAS described model verifications using QQplots. Models for which the QQplot diverges too soon from the bisector reflects an excess of false positives or negatives. In the first case, models should be enriched with covariates, while in the second some covariates may overcorrect the model. This latter case has been observed in several plant–virus GWAS, where models simultaneous corrected for kinship and the genetic structure [41]. For plant–virus GWAS that tested several statistical models, the best model was usually that correcting only for kinship (×4) or for both the genetic structure and kinship (×5) [72]. Gouy et al. (2015) tested both Q and Q + K models and selected the Q + K whereas QQplots showed overcorrection. They did not map any QTL [66]. For instance, models correcting only for kinship were found to be more accurate for mapping wheat resistance to BYDV and sugarcane resistance to multiple viruses [30,33,67].

LMMs require quantitative data and hypothesize Gaussian distributed residuals. Resistance to viruses is seldom quantitative (see Section 3.4.). Symptom severity scores (qualitative data) from plant repetitions are usually transformed into quantitative data via the calculated incidence, mean or through preliminary analysis of phenotypic data to extract the best linear unbiased estimators (BLUEs) or best linear unbiased predictors (BLUPs) of each genotype. BLUEs and BLUPs are usually highly correlated. Choudhury et al. (2019) tested the same GWAS model on BLUEs and BLUPs and mapped the same four QTLs, but the QTL significance levels were lower with BLUPs than with BLUEs. The hypothesis of Gaussian distribution of phenotypic residuals was assessed in only nine out the 42 using LMM, all using BLUPs as phenotypic data, and seven of them respected the hypothesis. Several GWAS models used in plant–virus GWAS were derived from LMM, including unified MLM, compressed MLM, Bayesian sparse LMM, FarmCPU, and latent factor mixed model (Appendix A).

### 4.2. Multilocus Models in GWAS: Detecting Markers Hidden by Major Effect QTLs

Virus resistance genes can be clustered in the plant genome, e.g., the Vat gene in melon [90], and unfortunately, LMMs are not able to distinguish colocalizing QTLs. Multi-loci LMM (MLMM) and multi-loci random LMM (MRLMM) are designed to overcome this issue [112,113]. MLMM is an iterative procedure that consists in adding at each step the most significative marker as a fixed effect to the current model. The resistance level is then corrected by the best genetic variant effect, and the next GWAS aims to associate the residual variability with the variability of other genetic variants. The model stops when no further genetic variants below the significance threshold are found. In principle, only the best SNP associated with each actual QTL should be kept. SNPs in high LD with those selected in the previous steps, will have their *p*-values rise above the significance threshold (i.e., a roundabout way to correct for the genetic structure effects for distant SNPs in LD). The significance of some peaks will sometimes increase during the steps, leading to detection of additional QTLs initially masked by QTLs with large effects [112]. This method can distinguish close loci that may have overlapped in a single peak in single step GWAS. MLMM was used in four plant–virus GWAS [43,72]. Tamisier et al. (2017) compared MLMM and LMM resulting from the same phenotypic dataset. Among the six SNPs that were significant with regard to PVY resistance, two were mapped by both models, three were only mapped by LMM (including two SNPs physically close to other significant SNPs), and one was only mapped by MLMM while having non-negligible effects.

### 4.3. Other Models That Are More Tailored to Virus Phenotype Distributions

In the plant–virus GWAS context, assuming that resistance traits have a normal distribution would mean that most individuals of the diversity panel would exhibit intermediate symptoms, with individuals with extreme phenotypes (susceptible or resistant) being rare. This scenario seldom occurs in the plant–virus context (Figure 3h,i) due to: (i) the presence of major resistance genes that are detrimental to the detection of intermediate resistance, and (ii) the rarity of resistant alleles leading to asymmetrical phenotype distributions, with an excess of resistant/susceptible individuals.

The absence of intermediate phenotypes and some plant virus resistance detections, e.g., the presence/absence of symptoms or viruses (qualitative ELISA or (RT)-PCR), produce **binary data**. Different GWAS models are adapted to this data distribution, e.g., case/control GWAS which have been commonly used to study the genetic control of human disease. In this case, the population is divided between healthy and infected individuals and Chi2 is performed to test for associations with genetic variants [114]. Resistance to the Wheat spindle streak mosaic virus (WSSMV) in wheat [58], to PPV in apricot [40] and to PPV in *A. thaliana* [28] was successfully mapped with this model. Surprisingly, in most plant–virus GWAS, binary data were considered to be quantitative and GWAS were run with a linear regression [47] or with an LMM [40]. Pagny et al. (2012) and Mariette et al. (2015) compared a Fisher exact test and an LMM for binary data. In both studies, the two approaches mainly detected different QTLs, despite relying on the same phenotypic and genetic data. In human medicine, LMM analysis of binary data is known to result in loss of power compared to analysis with logistic regression [115]. Logistic regression is adapted to binary data, but has never been used in plant–virus GWAS. So far, this model has only used by Chang et al. (2016) for regional associations (in contrast to genome-wide associations), to confirm QTLs detected by LMM [67]. They could, however, miss additional QTLs when other markers are not tested.

Alternatively, it is possible to normalize phenotypic data via statistical transformation, e.g., logarithm transformation, as described by Rubio et al. (2019), or Yeo–Jonhson transformation whereby asymmetrical data is transformed into normalized data, as described by Tamisier et al. (2020). GWAS mapping with normalized data results in loss of direct biological interpretation of estimated marker effects in post-GWAS analyses.

Other statistical models adapted to asymmetrical data distributions have been developed, but have never been used in plant–virus GWAS. Poisson models calculate the probability of observing each resistance level class and are expected to enhance the GWAS power for **ordered class data** such as the symptom severity score. They could also spotlight genes involved in different resistance levels, from susceptibility to intermediate resistance or from intermediate resistance to full resistance.

Putative allelic interactions should also be considered when implementing GWAS models. Indeed, many types of plant virus resistance are recessive [116] and therefore, require more complicated GWAS models to be detected. **Non-additive GWAS models** have been developed to test hybrid or landrace populations with substantial heterozygosis residuals [117]. Several sugarcane-virus GWAS relied on the R package polyGWAS which enables testing of different forms of genetic control, e.g., simplex or duplex dominant [35,65]. Although initially designed for polyploid species, for which allelic interactions are complex, this package can also be used for diploid species.

### 4.4. Describing QTLs: Significance, Number, Precision, and Allele Frequency

One key issue regarding GWAS is to be able to define the significance threshold for the *p*-values. In plant–virus GWAS models, the null hypothesis (H0) implies that there is no difference in resistance level according to the different allelic state. Usually, an α = 0.05 threshold is set to reject the H0 hypothesis. Markers with a *p*-value below the threshold are validated as being loci putatively involved in plant virus resistance. However, the risk of false positives increases with the number of tests. That is why multiple testing procedures are applied in order to account for the number of tests performed and control the Type I error rate at the selected nominal level. The Benjamini and Hochberg false discovery rate (FDR) [118,119] has been used by many authors of plant–virus GWAS, but the Bonferroni correction [120,121] was more popular. It consists of dividing α by the number of independent tests, i.e., by the number of genetic variants in the case of GWAS. The increased genetic variant number led to Bonferroni thresholds that were very selective for QTL detection. However, due to LD, SNPs are not totally independent, and consequently, their tests are also not independent. It is possible to divide α by the number of independent SNPs [122,123,124]. Conversely, it is also possible to perform GWAS only on independent SNPs and use a conventional Bonferroni threshold. Some authors preferred to arbitrarily set α, e.g., at 0.001 [63] or even lower [56] and QTLs were identified based on the peak shape [80,125,126].

GWAS results are usually presented on Manhattan plots displaying –log (*p*-values) of all SNPs at physical positions on the genome (Figure 7c). Manhattan plots provide, in one glance, different information on the genetic architecture of the plant virus resistance. SNPs with *p*-values under the significance threshold can be organized in peaks, due to LD, designing QTLs located in the corresponding genomic regions. Significant SNPs can also be isolated, reflecting either genomic areas with low marker coverage or false positives. However, when genetic markers were selected regarding the LD or tested with statistical models such as MLMM, single significant markers remained good QTL candidates. In that respect, Montes et al. (2021), who used FarmCPU to map *A. thaliana* resistance to CMV, detected eight QTLs, all supported by just one SNP.

The higher the peak is, the more robust is the QTL. However, the peak height is not directly correlated to the SNP effect (see 5.1.). Several methods are available to **delimit QTLs**: delimitation by the two SNPs flanking the SNP with the lowest *p*-value; delimitation by the two flanking SNPs containing all significant SNPs at the locus; and delimitation by the local LD block. The peak width depends on the LD decay in the vicinity of the causative gene. Narrow peaks are evidence of an ancient introduction or insufficient coverage, while wide peaks indicate QTL overlap due to clustered resistance genes, recent introduction, and selective sweep. In that perspective, GWAS might be a tool to investigate the QTL evolution process [127,128]. Successive Manhattan plots of each MLMM steps represent a useful tool to assess overlapped QTLs.

The **number of peaks and their location** at the genome scale provide a reliable picture of the genetic architecture of plant responses to viruses. A single peak suggests that resistance is controlled by one major gene, while several peaks suggest that the resistance is controlled by several genes. However, this latter point is less straightforward as several peaks can depict long-distance LD if the GWAS model has not accounted for the genetic structure. The number of peaks observed might be different depending on the statistical GWAS model implemented (see above and Section 4.1.). The number of QTLs identified in plant–virus GWAS ranges from 0 [66] to 31 [61].

Finally, the respective frequencies of each QTL allele can be compared to the MAF threshold. Indeed, QTLs with balanced frequencies are more reliable. The allelic frequency should also be studied within each genetic group. From a scientific standpoint, a higher frequency of a resistant allele in a specific genetic group than in others suggests a longer co-evolution. This could be an indirect way to understand the virus origin. Tamisier et al. (2020) described that PVY tolerance-associated alleles were significantly more frequent in genetic groups including genotypes from wild relative species than groups with cultivated genotypes. From the breeder’s viewpoint, donors can be selected in resistant groups to help the enrichment of susceptible groups.

## 5. The Post-GWAS Step: From QTL Validation for Breeding to Candidate Gene Research

### 5.1. Inferring QTL Effects and Identifying Favorable Alleles… A Complex Issue

Estimating the effects of QTLs detected by GWAS is not simple (Figure 8b). Indeed, when LMM are used, allele effects, i.e., one of the fixed effects, are hard to distinguish from random effects in the estimation process. QTL effects were mainly documented with an adjusted R^2^ in plant–virus GWAS, but without indicating the model used for this calculation. Montes et al. (2020) quantified the relative importance of SNP effects on *A. thaliana* resistance to CMV by calculating the mean square error difference for the predictions of two models, i.e., a complete model with the significant SNP and covariates and a model without this SNP. Rabbi et al. (2020) fitted a linear regression using all significant SNPs as a covariable and quantified allele effects of one SNP by substituting this SNP from the initial model. Finally, they tested the two effect differences with a pairwise t test to estimate QTL effects for cassava CMD resistance.

Alternatively, the effects of resistance QTL combinations can also be studied. By this approach, haplotypes combining QTL favorable alleles may be designed and the study may be geared towards a resistance ideotype construction. A combination of five alleles favorable for maize BYDV resistance explains up to 33.36% of the phenotypic variance [48]. It reflects additive effects or even epistatic effects if the combination of two or more favourable alleles is different from the sum of the individual allele effects. Aoun et al. (2020) successfully highlighted epistatic mechanisms for *A. thaliana* resistance to *Ralstonia solanacearum* [129]. Choudhury et al. (2019) provided boxplots showing variations in wheat BYDV resistance levels regarding the number of favourable alleles with the four mapped QTLs. Genotypes with one or two favourable alleles were not statistically different from those without favourable alleles, but genotypes with four favourable alleles were significantly more resistant than all other genotypes. The findings of analyses of QTL combinations can also be graphically displayed to highlight patterns in phenotypic variance partitions between significant markers, genetic backgrounds, and residuals at each step of an iterative MLMM GWAS.

The most significant SNPs sometimes explained a low percentage of phenotypic variance despite being closely associated with it. Actually, SNPs closely associated with barley BaYMV resistance, with –log10 (*p*-value) ~13, explained only 8% of the phenotypic variance [60]. Studies mapping QTLs with more than 10–20% effect have been rare, while combination of several QTLs seldom explains more than 30%. Therefore, most phenotypic variance is not explained and is referred to the ‘missing heritability’; this is the cornerstone of GWAS criticism in the human genetics research [130,131]. Several factors could explain this phenomenon, some of which have been discussed in the pre-GWAS step (too small populations, unprecise phenotyping, rare SNPs, rare phenotypes, genetic heterogeneity, ill-adapted statistical models), to which we could add: (i) the complexity of the genetic architecture; (ii) epistatic interaction genes; and (iii) epigenetic variation [132,133,134,135]. The local score approach was proposed to map non-significant low effect QTLs [135,136]. However, local score studies have yet to be used to for QTL detection in plant–virus GWAS.

### 5.2. Inviting QTLs in Breeding Programs

In successful scenarios, QTLs mapped by plant–virus GWAS were converted into breeding markers. A preliminary step is often to identify robust QTLs, that are of the most interest for breeders. Two strategies may be deployed: (i) validate QTLs at the genetic level; or (ii) at the environment level.

In most plant–virus GWAS, at least one QTL co-localized with a known resistance QTL mapped by segregating population analysis, thereby a priori validating the method (Figure 8a). For instance, 15 significant SNPs mapped by *A. thaliana*-PPV GWAS co-localized with the *sha3* gene, which was previously mapped for the same resistance [28]. Post-GWAS RIL building can be carried out to target GWAS QTLs and enhance the mapping precision, where only progeny which are heterozygous at the QTL and fixed susceptible for the rest of the genome are selfed. GWAS panels consisting of elite lines derived from few founder lines can have long LD blocks. In this case, a solution would be to enrich the panel with lines that are more genetically diverse in this locus and rerun GWAS [5].

QTLs for plant virus resistance are considered to be robust if only they occur in any phenotyping environment [73] (Figure 8c). Indeed, GxE interactions for plant virus resistance are not negligible (see Section 3.4.). Hao et al. (2015) tested the same GWAS population for three years and mapped three different QTL sets [5]. Two strategies were proposed to analyze these datasets resulting from different trials. In the first one, all phenotypes from the different datasets were combined in one large dataset, and the GWAS model was corrected with environmental covariates (**GigaGWAS**), while environmental effects were calculated with the shared genotypes. The alternative would be to perform a single GWAS on each dataset and combine *p*-value datasets (**MetaGWAS**). Kayondo et al. (2018) tested both strategies for cassava resistance to CBSV. For symptom severity on leaves, GigaGWAS could successfully map QTLs similar to those of one of the single-environment GWAS studied in MetaGWAS. Conversely, for symptom severity on roots, GigaGWAS did not map any QTLs, contrary to one of the single-environment GWAS. The authors concluded that GxE interactions for root severity could have been too substantial.

Multitrait information, e.g., vector–virus relationships or viral diseases involving several viruses, is also of prime interest for breeders. A nice example concerns a Maize lethal necrose disease (MNLD) study resulting from the co-infection by MCMV and SCMV [46]. The maize response to both viruses were independently studied by two different teams (Table 1) and further co-analyzed. Nyaga et al. (2020) mapped 32 significant SNPs for maize MNLD resistance, one of which was close to a QTL involved in maize SCMV resistance, while another one was in the confidence interval of a QTL for maize MCMV resistance alone, and all of the other were associated with both MCMV and MLND resistance [46]. This study did not find any new QTLs, but helped to reduce former QTL widths. QTLs mapped by GWAS could also be compared to QTLs of resistance to other viruses in the same plant species, putatively highlighting broad-spectrum QTLs. GWAS models taking into account biological covariates, such as vector attractivity or infection with other viruses, could enhance the GWAS power for mapping new QTLs [29].

Finally, QTLs with a major effect could be converted into **breeding markers for marker assisted selection (MAS)**. Hourcade et al. (2019) converted the main QTL mapped by wheat-WSSMV GWAS into a KASP marker [58]. Minor effect QTLs are essential for the resistance durability and good major gene performance. Indeed, major resistance alleles introduced in intermediate backgrounds would be more durable than the same major resistance alleles introduced in a susceptible background [43]. This scenario was demonstrated by the *pvr2^3^* gene conferring resistance to the PVY introduced in two pepper backgrounds [16]. Eighteen QTLs were mapped for barley BaMMV and BaYMV resistance that explained up to 32% of the phenotypic variance [60]. These 18 QTLs would be hard to mainstream in conventional breeding programs. **Genome-wide selection (GWS)** is becoming increasingly popular and is an interesting alternative to MAS in case of multiple QTLs with minor effects. GWS rely on the same population and datasets as GWAS. In that perspective, pre-GWAS issues are preserved [29,33,45,72], but the statistical objectives fundamentally differ: GWAS aims to map resistance while GWS is geared towards predicting the resistance level of a genotype based on its allele combinations. This method significantly improved the level of cacao resistance to a fungus while GWAS could not map any QTLs [137]. In several plant–virus GWAS publications, GWS was used in the analyses while obtaining similar results, especially in breeding for resistance to MLND due to the complex resistance architecture [29]. GWAS and GWS are complementary approaches. Indeed, GWS models enhanced by QTL mapped in GWAS could be more accurate than naive GWS models when the detected QTLs have major effects [73].

### 5.3. From QTLs to Causal Genes, a Budding Approach

All genes within a local LD block containing a QTL should be considered (Figure 8a). Indeed, causal mutations could have poor associations due to genotyping issues. For instance, if there are no SNPs located near the causal locus, GWAS will detect the locus in strong LD with the causal locus but will not detect the closest locus. It is possible to shorten the candidate list to genes with non-synonymous SNPs in exon sequences (Figure 8d). However, in 2012, in human GWAS studies, 90% of disease-associated SNPs were located in non-coding sequences [138]. Candidate gene research relies on the annotation quality of the reference genome [67]. The number of genes detected in plant virus GWAS ranges from 0 candidate gene to 2605 candidate genes [58]. Three scenarios can be described once the candidate gene list is established.

First, the candidate gene list contains some genes already known to be involved in the virus biological cycle (susceptibility genes) or to trigger plant resistance (resistance genes). Several QTLs detected by GWAS co-localized with a known resistance, discovered via molecular studies. Hao et al. (2015) highlighted the GLE1-like protein involved in RNA movement to cytoplasm as a candidate gene. Candidate genes involved in maize hypersensitive reactions to MCMV were mapped by Gowda et al. (2015). A hypersensitive response is considered to prevent virus propagation from initially infected to neighboring cells, and virus entrance in sieve tubes. Systemic resistance, with genes such as those in the restricted tobacco etch virus (TEV) movement (RTM) gene family, restrict long-distance virus movement in plants, e.g., TEV and PPV long-distance movement in *A. thaliana* [28,139].

Second (and in most GWAS), GWAS detected numerous unknown genes or genes not known to be involved in resistance mechanisms. In this scenario, all genes should be independently studied and prioritized regarding an argued and crossed selection (databases, literature, transcriptomic studies, haplotype studies, gene enrichment, etc.).

Third, some genes expected to be involved in resistance are not found, for instance, because they have no allelic diversity within the diversity panel. Perfect examples of this scenario are the eIF4E and eIF4G gene families, which are involved in several types of plant potyvirus resistance [140]. Other eIF4 genes were candidates in plant–virus GWAS, for instance, for two bymoviruses (BaYMV and BaMMV) in barley [60] and for one machlovirus (MCMV) in maize [45]. Other genes involved in plant–virus resistance have never been mapped in plant–virus GWAS, such as the vacuolar protein sorting 4 involved in cucumber potyvirus resistance [22].

All plant–virus GWAS focused lists of candidate genes based on the gene function rather than on the mutation effects and only described genes involved in known resistance mechanisms or plant growth. To our knowledge, QTLs mapped by GWAS for plant virus resistance have been functionally validated only once [41], i.e., for resistance to TuMV in Chinese cabbage with mutants from the related species *A. thaliana*. Candidate gene validation is hence a long and complicated process. Sometimes, gene validation was not described in the same article as GWAS, but rather in a subsequent article and is therefore not referred to in this review.

## 6. Conclusions

Most plant–virus GWAS were able to re-map previously known QTLs, thus showcasing the efficiency of GWAS mapping in the plant virus resistance context. When previous QTLs were not mapped by GWAS, it was hypothesized that the mutation involved was not frequent enough to reach the MAF threshold. If so, the resistance should be mapped by a segregating population rather than GWAS [80,141]. New QTLs were also mapped, suggesting that GWAS could map unknown resistance genes. In the future, innovative statistical models could be implemented to boost the potential of this strategy.

GWAS initially require considerable effort for choosing and genotyping the diversity panel. However, the investment returns are fast as the population can be used to phenotype and map a multitude of various traits. In this perspective, building the population regarding several viruses or several strains from the same virus should be considered to detect broad-spectrum QTLs. Moreover, the population could be phenotyped for any new emerging or re-emerging viruses. In this case, there is a higher probability of finding a relevant population subset to map new resistances by GWAS when a large panel is used. Collaborative projects involving public research and breeding companies are required to assemble these large panels pooling accessions from wild, cultivars, and elite lines.

Plant breeding for virus resistance is crucial for the sustainability of many crops. GWAS provide scientists and breeders with a major opportunity to accelerate the process, increase the genetic diversity of resistance, and to gain greater insight into quantitative resistance (so-called background effect). GWAS is thus expected to improve resistance durability.

To take full advantage of GWAS opportunities, joint association analyses of the genetic variability in plants and viruses should be a joint objective. Wang et al. (2018) developed the analysis with a two-organism mixed model (ATOMM) and tested it on a dataset collected via the inoculation of 130 *A. thaliana* lines with 22 *Xanthomonas arboricolae* strains [142]. First, they ran two marginal GWAS, on the plant and virus genomes, and then they ran a joint GWAS. Interestingly, they found that most of the heritability (44%) was supported by the *X.*
*arboricolae* genotype, while the *A. thaliana* genotype explained only 2% of the heritability. They implemented several options required for plant virus interaction studies, which have been described in this review: Gaussian or binomial models, genetic structure correction (i.e., the plant and virus genetic structures), as well as multiple variant genetic matrixes (multiallelic SNP, InDels). The joint GWAS detected significant QTLs that differed from those detected by marginal GWAS. Manhattan plots are now shifting from 2D (–log10 (*p*-value) × plant genome) to 3D (–log10 (*p*-value) × plant genome × virus genome) displays. This is probably a good strategy to identify genes involved in plant–virus relationships, yet it has so far gone under the radar. The analyses require a huge phenotyping effort, which only collaboration between virology and plant genetics teams could manage.

## Figures and Tables

**Figure 1 cells-10-03080-f001:**
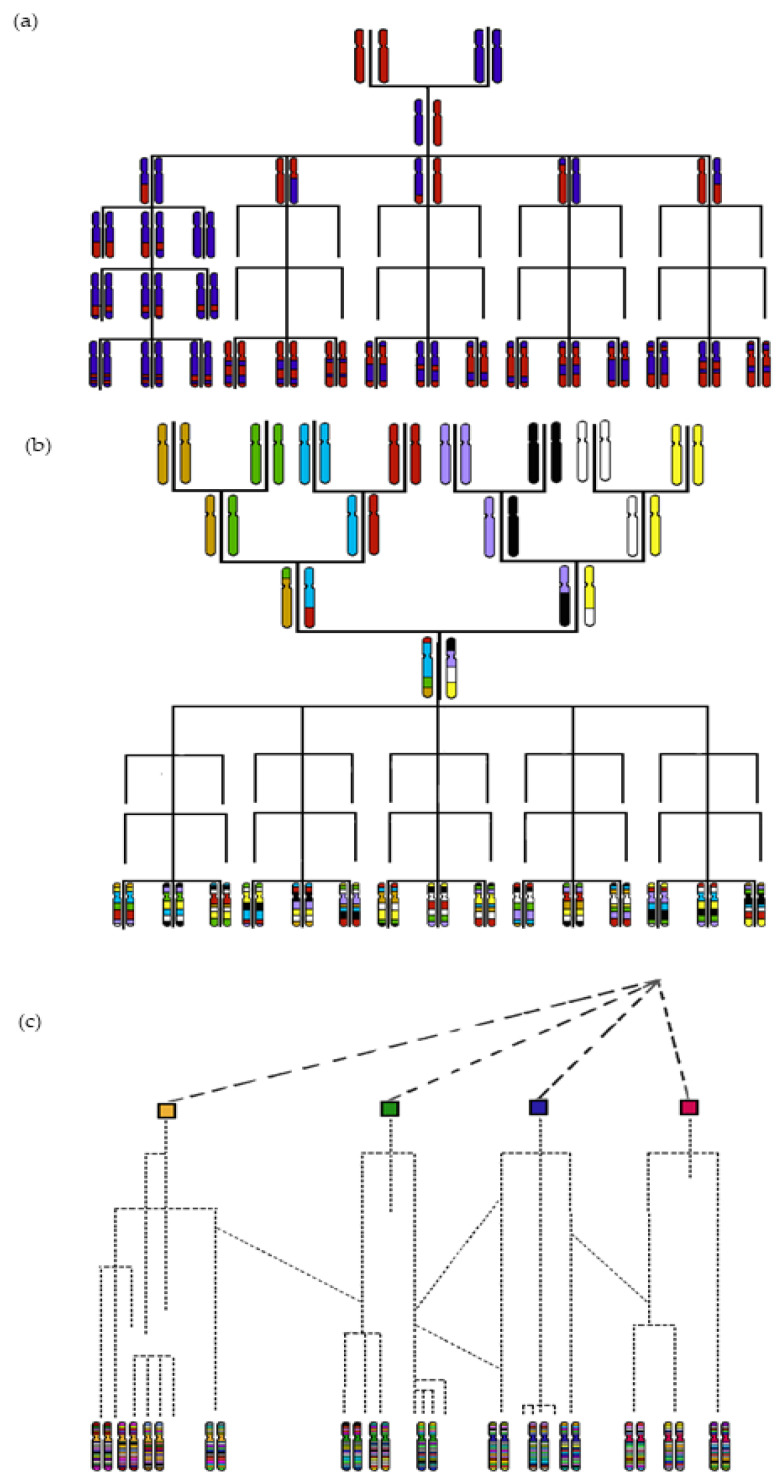
Linkage disequilibrium (LD) bloc exchanges among generations in different populations (**a**) Recombinant inbred line population. The allelic diversity studied is limited to the genetic variation between the two parents. LD bloc extents decrease with the number of generation. (**b**) Multi-parent advanced generation inter cross (MAGIC) population, contrasted parents are crossed to obtain heterozygous individuals, which are thus selfed by single seed descenton several generations to obtain the MAGIC population. The allelic diversity is limited to the initial parents. (**c**) Diversity panel, allelic diversity, and mapping precision result from the history of the species. Here, we illustrated the case of a population resulting from four fictitious ancestors.

**Figure 2 cells-10-03080-f002:**
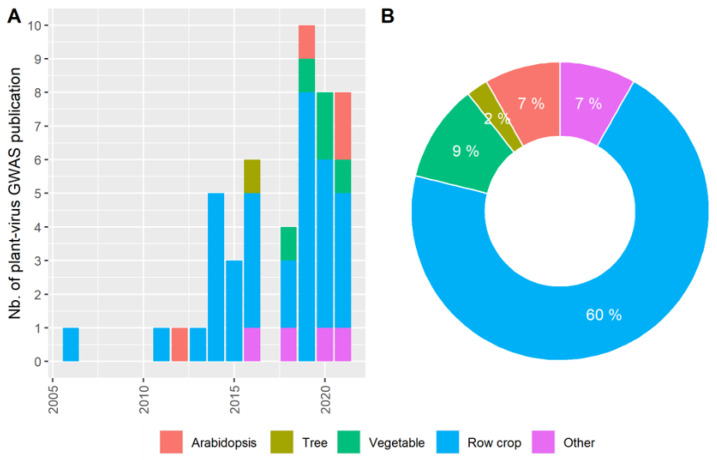
Overview of the publications of plant–virus GWAS: (**A**) Number of plant–virus GWAS articles published each year; (**B**) Proportion of plant–virus GWAS articles regarding the crop type. “Other” contains publications on peanut, cassava, and yam.

**Figure 3 cells-10-03080-f003:**
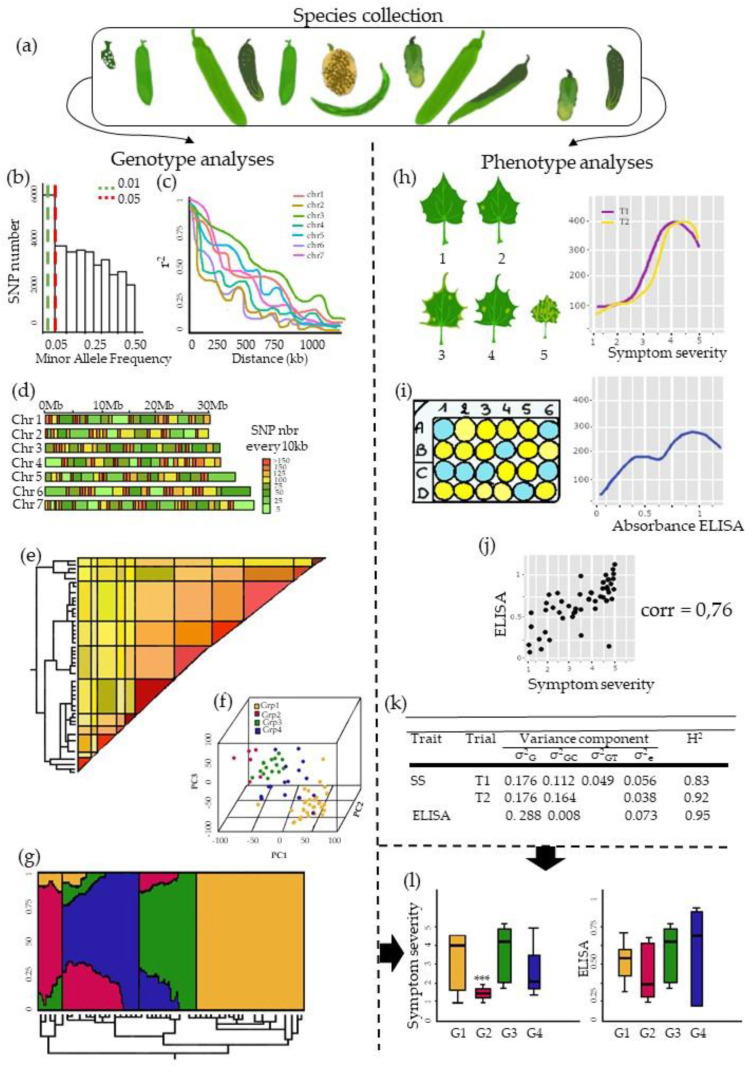
Overview of the main figures of a fictitious pre-GWAS pipeline. (**a**) A collection of genotypes is genotyped and phenotyped. (**b**) Histogram of minor allele frequency distribution with thresholds 0.01 (green) and 0.05 (red). (**c**) Genome-wide linkage disequilibrium decay for the seven chromosomes. (**d**) The density of SNPs on the seven chromosomes within 10 kb window. (**e**) A heatmap of the kinship matrix suggested high level of relatedness among some genotypes. (**f**) Principal component analysis indicates the structuration of the population in at least three groups. (**g**) Population structure; four subpopulations were identified, red, blue, green. and yellow. (**h**) Genotypes are phenotyped regarding a symptom severity (SS) scale ranging from 1 (no symptom) to 5 (leaf deformation and severe yellowing) two consecutive years (T1 and T2); the SS distribution displays an excess of susceptible genotypes. (**i**) The viral load is approximated for T1 via ELISA absorbance, the absorbance distribution follows a bimodal distribution, and highly resistant genotypes are rare. (**j**) The correlation between SS and ELISA absorbance phenotypes is high. (**k**) Heritabilities for SS for T1 and T2 and ELISA absorbance are great. (**l**) SS T1 and ELISA absorbance of the four genetic subgroups; the second genetic subgroup is significantly more resistant for SS T1.

**Figure 4 cells-10-03080-f004:**
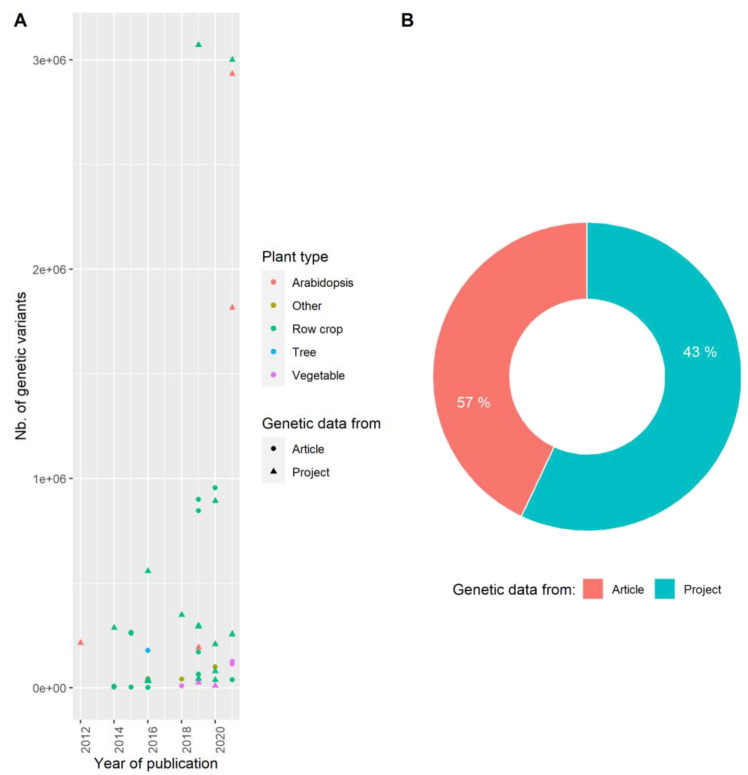
The genotypic variants in plant–virus GWAS: (**A**) Number of genetic variants used to map plant resistance to virus via GWAS in studies published since 2012; (**B**) Proportion of plant–virus GWAS articles which genotyped the diversity panel versus articles using already available genotypic data.

**Figure 5 cells-10-03080-f005:**
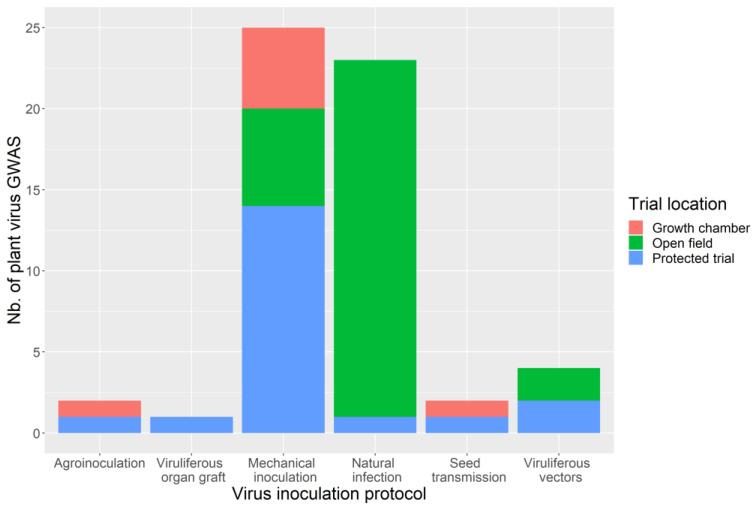
Number of plant virus GWAS regarding virus inoculation protocol and trial location. ‘Mechanical inoculation’ pools inoculations by hand rubbing and mist blower. ‘Protected trial’ pools plant virus GWAS trials which took place in greenhouses, glasshouses, and net protected fields.

**Figure 6 cells-10-03080-f006:**
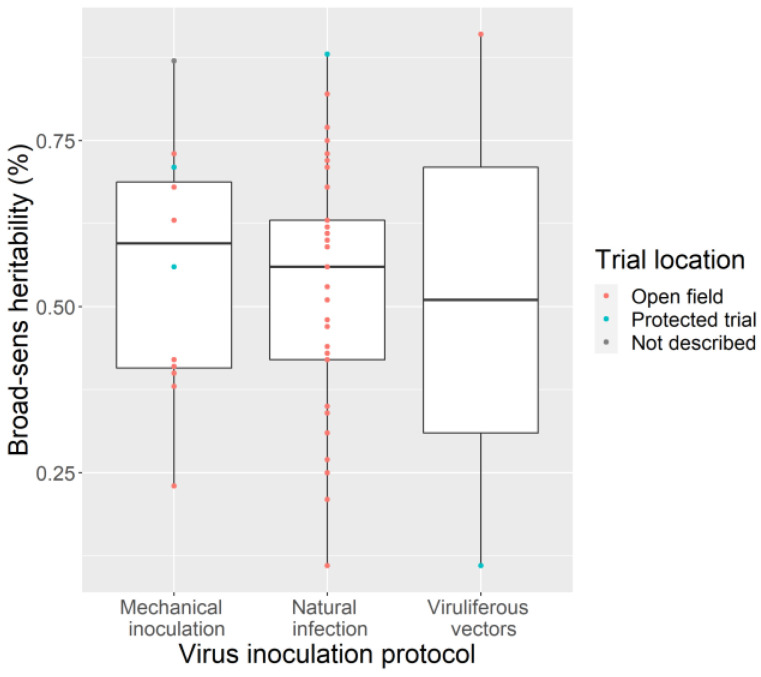
Broad-sense heritability calculated from 47 experiments conducted in 12 plant–virus GWAS (Appendix A) using the symptom severity as phenotyping trait regarding the inoculation protocol and the trial type. ‘Mechanical inoculation’ pools inoculations by hand rubbing and mist blower. ‘Protected trial’ pools plant virus GWAS trials which took place in greenhouses, glasshouses, and net protected fields. The second phenotyping step involves collection of data documenting the different resistance levels. Hereafter, the terms of ‘qualitative’ and ‘quantitative’ are associated with the description of these phenotypic data. Qualitative phenotypes refer to resistance data distributed in classes while quantitative phenotypes refer to resistance data exhibiting continuous values.

**Figure 7 cells-10-03080-f007:**
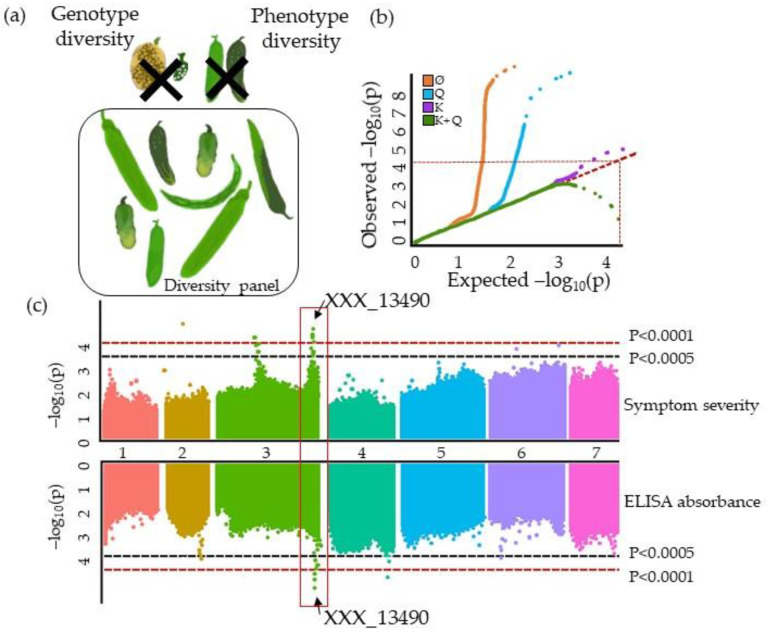
Overview of fictitious GWAS main figures, following pre-GWAS presented in Figure 3 (**a**) genotypes with low quality genetic data and/or too genetically isolated and/or too genetically close from other genotypes or genotypes displaying seldom phenotypes are discarded from the GWAS population. (**b**) QQ-plots for SS T1 for models without covariates (orange) and Q model (blue) present an excess of false positives. Most *p*-values of the QQ-plot of the K model (purple) are similar to the expected bisector, indicating K model appropriateness. QQ-plot of the Q + K model (green) presents an excess of false negatives and suggests overcorrections. (**c**) Manhattan plots for the two traits, SS above and ELISA absorbance below. The red and the black dashed lines indicate Bonferroni-corrected thresholds. A single QTL exceeds the significance threshold for both traits in the end of chromosome 3. SS displays another peak on the center of chromosome 3, with XXX_13490 being the top SNP, and three single SNPs on chromosome 2 and 6. ELISA absorbance displays three other peaks, respectively on chromosome 2, 4, and 7.

**Figure 8 cells-10-03080-f008:**
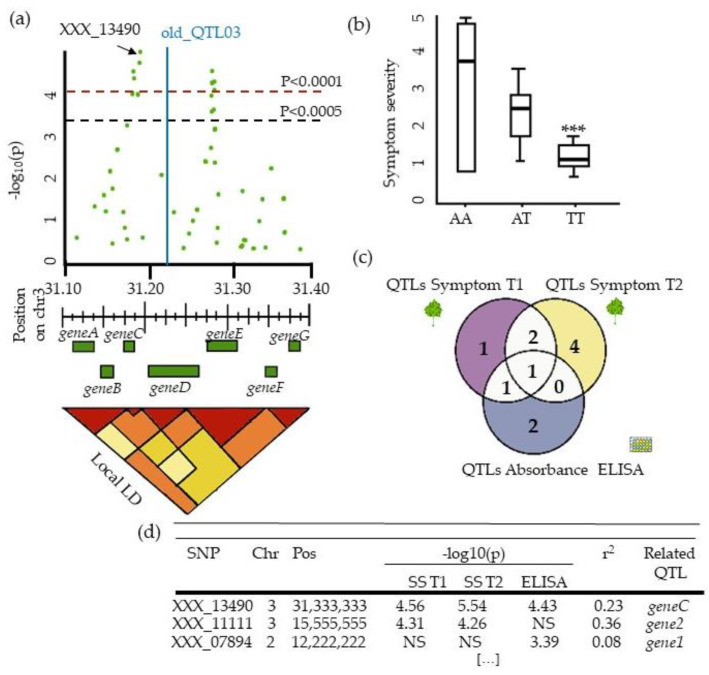
Overview of the main figures of a fictitious post-GWAS pipeline following the pre-GWAS and GWAS presented in Figure 3 and Figure 7 (**a**) Regional plot of SS T1 zoomed on the QTL in the end of chromosome 3 and colocating with ELISA absorbance peak. The most significant SNP is XXX_13490colocalizing with the historical old_QTL3 that was mapped with RILs. The geneC is the single gene in its local linkage disequilibrium bloc. (**b**) Analysis of the effect of the XXX_13490 alleles of SS T1, (***) TT genotypes are significantly more resistant than AA and AT genotypes (**c**) Venn diagram showing the number of unique and shared QTLs between SS T1, SS T2, and ELISA absorbance T1. (**d**) Effect of significant genes detected using GWAS for the three traits.

**Table 1 cells-10-03080-t001:** Overview of the pathosystems covered by the 48 plant–virus GWAS publications. In bold are the names of viruses studied in several plant species. Bracketed numbers represent publications in which the same plant–virus pair was studied. Further details on the plants, viruses, protocols, and GWAS models and results are outlined in Appendix A.

Species	Latin Name	Virus	Article
	*Arabidopsis thaliana*	Cucumber mosaic virus (CMV) **Turnip mosaic virus (TuMV) (×2)** **Plum pox virus (PPV)**	[28,37,38,39]
Apricot	*Prunus armeniaca*	**PPV**	[40]
Chinese cabbage	*Brassica rapa*	**TuMV**	[41]
Pepper	*Capsicum annuum*	Potato virus Y (PVY)	[42,43]
Common bean	*Phaseolus vulgaris*	Bean golden yellow mosaic virus (BGYMV)	[44]
Watermelon	*Citrullus lanatus*	Papaya ringspot virus (PRSV)	[36]
Maize	*Zea maize spp Mays*	Sugarcane mosaic virus (SCMV) **(×5)** Maize chlorotic mottle vírus (MCMV) SCMV + MCMV (MCMD) **(×3)** Mal de Rio cuarto virus (MRCV) Rice black-streaked dwarf virus (RBSDV) (×2) **Barley yellow dwarf virus (BYDV)**	[5,29,45,46,47,48,49,50,51,52,53,54]
Oat	*Avena sativa* L.	**BYDV**	[55]
Wheat	*Triticum aestivum*	**BYDV ** Soil-borne wheat mosaic virus (SBWMV) **(×2)** Wheat spindle streak mosaic virus (WSSMV)	[30,56,57,58]
Barley	*Hordeum* sp.	Barley mild mosaic virus (BaMMV) **(×2)** Barley yellow mosaic virus (BaYMV) **(×2)**	[59,60]
Rice	*Oryza sativa* L. *Oryza glaberrima*	Rice yellow mottle vírus (RYMV) **RBSDV (×3)**	[6,34,61,62]
Sugarcane	*Saccharum officinarum Saccharum spontaneum*	Fiji disease virus Sugarcane yellow leaf virus (SCYLV) **(×4)**	[35,63,64,65,66]
Soybean	*Glycine max*	Tomato ringspot virus (TRSV) Peanut mottle virus (PMV) Bean pod mosaic virus (BPMV) Soybean mosaic virus (SMV) **(×3)**	[33,67,68,69]
Peanut	*Arachis hypogaea* L.	Tomato spotted wilt virus (TSWV)	[70]
Cassava	*Manihot esculenta*	Cassava mosaic disease (CMD) **(×2)** Cassava brown streak virus (CBSV)	[71,72,73]
Yam	*Dioscorea rotundata*	Yam mosaic virus (YMV)	[74]

## Data Availability

Not applicable.

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
