# Peer review of "Deciphering the Genetic Architecture of Plant Virus Resistance by GWAS, State of the Art and Potential Advances"

_cells, 2021, doi:10.3390/cells10113080_

Round 1
Reviewer 1 Report
The manuscript by Mannot et al. is an important contribution in the field of GWAS related to plant virus resistance. The authors have summarized all findings till date very well and have given their insights for candidate gene prediction and their future validation. I strongly recommend the article for publication; however, authors should also summarize at the end somewhere how selective sweep can help to further deepen our understanding in this area. Authors can take examples from other studies where this approach has been used along with GWAS to dissect traits and to clearly identify signature of selection.
Reviewer 2 Report
The current manuscript presents new information, but the way the information is packaged and presented probably needs to be changed to accentuate the important points on ''Plant viral GWAS'' studies and require a more mature and focused presentation.

Round 2
Reviewer 2 Report
The manuscript has improved since the authors rewrote some captions for figures correctly. However, some minor issues still remain.
- The caption in Figure 1 mentioned “Linkage bloc, LD bloc” and should mention the correct words.
- L82 When the authors used abbreviation forms (MAGIC and SSD), I recommend using long forms first in the paper.
- provide more clear scales and indicators for axis X and Y of figure 5 and figure 6.
Author Response
We carefully considered all new suggestions from reviewer 2:
- We harmonized the figure caption with “Linkage disequilibrium (LD) blocs.
- We switched from abbreviation forms to long forms for Linkage disequilibrium (LD), Recombinant inbred line (RIL), single seed descent (SSD) and Multi-parent advanced generation inter cross (MAGIC)
- We reprinted figure 2, 4, 5 and 6 with more annotated scales (both x and y axis). In addition, we completed figure captions for some general items (cf mechanical inoculation and protected trials).